# The right to water: Impact on the quality of life of rural workers in a settlement of the Landless Workers Movement, Brazil

**Priscila Neves-Silva** [ID]*, **Juliana Aurora de Oliveira Lopes, Léo Heller**

Department of Public Policies and Human Rights for Health and Sanitation, Rene Rachou Institute, Oswaldo Cruz Foundation, Belo Horizonte, Minas Gerais, Brazil

* priscila.neves31@gmail.com

## Abstract

Access to water for rural populations is vital not only for personal consumption and hygiene but also for food production, income generation and cultural practices. To deepening the understanding of this issue, this research addressed the access to water in a settlement of the Landless Workers Movement. The perspective of the Human Rights to Water and Sanitation was used as a theoretical framework, assessing how inadequate access to water impacts the quality of rural populations. A qualitative research was used, through participant observation and individual interviews with 12 rural workers, living at the Ulisses Oliveira settlement. The findings reflect that water is not sufficiently available to meet the community's social, economic and cultural needs and that such conditions can lead to a loss of identity. Therefore, access to water must be understood in the light of its political, social and cultural dimensions and the Human Rights to Water and Sanitation can be used as an instrument to public policies.

## Introduction

The human rights to water and sanitation (HRWS), explicitly recognized in 2010 by the UN General Assembly and the Human Rights Council, opens up a possibility to operate as a framework to regulate the access to water and have the potential of significantly contributing to public policy formulation and implementation [1]. In this framework, access to water must be accessible, affordable, safe, acceptable and available in sufficient quantities for personal and domestic use [2]. In addition to these aspects related to access to water, sanitation services must ensure hygiene, the privacy of access and dignity [3].

In this way, it can be reaffirmed that all people, without discrimination, need access to water and sanitation services that are not detrimental to health. Furthermore, recognizing access to water and sanitation as a matter of human rights means that countries must ensure basic principles such as equality and non-discrimination, accountability, transparency, access to information and participation [2].

Acknowledging water as a human right must also entail recognizing that people's needs for water can be different from one community to another, given the uniqueness of culture and lifestyles. For rural communities, water usually embodies not only a material but a symbolic dimension. Essential not only for drinking and ensuring personal and domestic hygiene, water

preserve the participants' anonymity. Data are available from the Rene Rachou Institutional Data Access (cepcoord.minas@fiocruz.br) for researchers who meet the criteria for access to confidential data.

**Funding:** Oswaldo Cruz Foundation founded Priscila Neves Silva with a scholarship but it did not play any role in the study design, data collection and analysis, decision to publish, or preparation of the manuscript.

**Competing interests:** The authors have declared that no competing interests exist.

is vital for food production, which is an essential aspect for this demographic's food sovereignty and activities associated with their cultural practices [4–10]. However, families in situations of poverty that live in rural areas have 29% fewer chances of having access to improved water sources compared to their counterparts in urban areas [4]. In this way, it can be said that deficiency in access is defined in different ways according to each community, owing to which accessibility requires a context-specific analysis in each case.

Some authors adopt a more limited vision of the HRWS. Van Koppen *et al.* [11] argue that its scope is limited to personal and domestic use, not encompassing the reality of demographics living in rural regions and their multiple water-related needs. For Goff and Crow [7], the HRWS mainly focus on bio-medical health aspects, not including matters related to different demographics' lifestyles. Jepson *et al.* [12] argue that the HRWS is focused on western culture, which treats water as an object and does not apprehend it through a relational perspective with political and cultural dimensions. For these authors, this more restrictive definition of the HRWS only considers drinking water and does not commit to acknowledging the importance of access for food production, in addition to other social, political and cultural needs.

On the other hand, Metha [6] adopts a broader view, arguing that the HRWS, which was already a right enshrined in the International Convention on Economic, Social and Cultural Rights, encompass water for other uses including productive ones, as well as culturally-specific matters. Recently, the right to water for food production, at a small scale, has an important recognition by the UN United Nations Declaration on the Rights of Peasants and Other People Working in Rural [7]. Through these interpretations, the right to life is also the right to the means of subsistence and water policies cannot limit the use of water only for personal and domestic uses. In this respect, it would be impossible to separate, in rural areas, household uses of water from productive ones, as residents are not able to prepare food, which is a household use of water, if they do not have food production. Therefore, a more multidimensional vision of access to water is needed that takes communities' autonomy, identity and freedom into account [6]. This logic requires the HRWS not to be limited to prescriptive standards when defining what constitutes the enjoyment of these rights; their scope must consider the different needs of each community. Ensuring access to water under this broader framework allows that communities use water in the way that most suits their needs, traditions and lifestyles, ensuring the quality of life and equality in access [6]. It is important to remark that the HRWS do not affirm that water should be free of charge, but that affordability should be ensured and that tangible and intangible costs to access water should not impact the enjoyment of other social and economic rights. Therefore, the understanding that access to water is context-specific, depending on the needs and values of each community, gives another frame for interpreting the meaning of water accessibility.

In addition, other authors have discussed the relationship between food sovereignty and the human rights framework with a view to committing governments, through binding legal mechanisms, to ensure the interests of the rural population and to attribute priority to their use of water. Doing so would ensure that people's basic needs are covered and would avoid water allocation from being influenced by power relations [13, 14]. In line with this discussion, Via Campesina pointed out that the right to water must be understood within the framework of food sovereignty, as food sovereignty is related to agroecological practices and small scale production, contributing to the protection and sustainable use of the water. For them, local and traditional knowledge about water management needs to be considered in order to have an ecosystem protected and to ensure food sovereignty [15].

MST emerged in the 1980s as the result of grass-roots struggles for land. The movement aims to occupy non-productive land to allow rural workers the possibility to produce food, thereby generating income and food for their own consumption, with the intention of achieve food sovereignty. During their occupation of land families encamp until gaining possession of

the land, at which point they establish a settlement and acquire the right to use the land. Currently, the movement is present in 24 of Brazil's states and, according to the latest estimates, is composed of 350,000 settled families and 120,000 camped families [16].

In this context, this research is guided by the HRWS framework, for understanding the access to water by the community that lives in a settlement of rural workers managed by the Landless Workers Movement (Movimento dos Trabalhadores Rurais Sem Terra–MST) in Brazil. It also aimed to understand how the lack of sufficient water thereof impacts on different dimensions of the lives of the residents. For this analysis, the HRWS framework was applied not only through their normative content, but basic principles of human rights, as non-discrimination and participation, were also considered, taking into account multiple needs of the community and valuing people's voices and knowledge,

## Materials and methods

The findings presented herein are drawn from a qualitative study carried out at a settlement of the Landless Workers Movement at Vale do Rio Doce, Minas Gerais, Brazil, called Ulisses Oliveira. The research comprised two main elements: (i) participant observation investigating access to water and sanitation in the settlement, and (ii) twelve interviews with inhabitants of the Ulisses Oliveira settlement.

### Data collection

The Ulisses Oliveira settlement was chosen as the location to carry out the present research. The settlement, founded in 2005, is the fruit of MST occupations in the region of Vale do Rio Doce (Doce River Valley) since 1989. With an area of 1,463 hectares, the settlement is occupied by 44 settled families.

In October 2017, two qualitative researchers spent one week as guests of a family living in the settlement, the first author (PNS), who has a PhD in Public Health, with a focus in Human and Social Science in Health, and is working as postdoctorate researcher at Fundação Oswaldo Cruz, with a great experience with qualitative research methodologies, and the second author (JL), who has a Master degree in Public Health, with a focus on Human and Social Science in Health. During this period, the researchers observed how families in the settlement access water for personal use and domestic activities, as well as their sanitation practices. In addition, the researchers observed how they use water for agricultural production, as farming and livestock production are the main income generating activities.

All observations were recorded in a research diary and analysed accordingly. The participant observation phase was essential to gain a better understanding of the ways the families use water and sanitation services [17].

Individual interviews were also used to gather data [16]. These interviews followed a semi-structured script, elaborated based on the researchers' experience and a literature review. The script contained questions seeking general information about the history and everyday life of the settlement, the normative content of the HRWS (availability, quality and safety, accessibility, affordability and acceptability) and issues reflecting some of the human rights principles (non-discrimination, access to information and transparency, and active and meaningful participation). The interviews were carried out by the first author. A pilot test to validate the script was performed one month before, with a member of the community that was in Belo Horizonte for a MST meeting. Men and women, all rural workers, older than 18 years of age, living in the settlement, were invited to participate in the study. Twelve people were interviewed individually, they have from 28 to 76 yaers old, seven were women and five men. The interviews were carried out in the individuals' houses in the settlement.

Participants were selected according to the criteria of accessibility and snowball sampling [17, 18]. The family where the researchers were hosted indicated another family to participate in the research. That family, in turn, indicated another one and, in this way, the total number of participants was attained. The researchers were, however, mindful of interviewing families living in different areas of the settlement. The saturation sampling technique, in which the absence of new themes and repetition of content in interviews are indicative that the main ideas have already been raised, was used to define the number of interviewees [19].

Participants were informed about the project's aims and were invited to participate voluntarily and to sign an Informed Consent Form. The participants were advised that they could feel free to leave the study at any time.

## Data processing and analysis

The interviews were recorded, transcribed and analyzed by both qualitative researchers (PNS and JL).The content analysis technique was used, through which the information collected was systematized into thematic categories [20]. This involved reading through participants' responses and assigning codes to specific aspects of their responses.

In order to reduce bias and the influence of the researchers' respective background during the analytical process it was done by both researchers individually. Both possessed a research diary, performed data analysis independently and then compared their findings.

Both researchers separately read through all the transcriptions and drafted a list of recurrent codes derived from the data. The codes were defined independently and then refined collaboratively to ensure adequate intercoder reliability. The coders discussed discrepancies between their codes until they reached consensus. A code book was then developed, discussed and accepted by the researchers. A list of key themes, based on the frequency with which each code was identified throughout the interviews, was then created: Availability: multiplier effect compromising the HRWS; and Water as an income generation and subsistence tool. Respondent statements that are particularly expressive, informative and representative were selected and are presented herein as direct citations. Illustrative quotations were reviewed by the researchers for consensus.

## Ethical considerations

The collected data is confidential and the anonymity of the participants has been guaranteed; the names of participants will not be disclosed under any circumstances. Participants that were individually interviewed were identified with the abbreviation LMF (Landless Movement Family) followed by the serial number in which they were interviewed. The survey was approved by the Research Ethics Committee of the René-Rachou Research Center under protocol CAAE 49209515.0.0000.50.91.

## Results

### Unavailability: A catalyst for violations of the HRWS

The period in which the research took place is a dry season in Brazil. Accordingly, the participants reported that the quantity of water available in the settlement had been unsatisfactory throughout the entire year, entailing losses in production. While the research was taking place, every family was in search of a way to obtain access to water. Some families had springs or dug wells in their plots. Those that did not have such resources collected water from their neighbours' springs or wells. Others fetched water from a school located within the settlement, which obtained water from a nearby spring. One family fetched water from a small stream near its house.

A well was built close to the settlement's school as a source of water for the community. However, the well was closed for many years and the population did not have access to it. Now, according to the inhabitants' accounts, it cannot be used due to the high concentration of metals, including iron.

> "It has a lot of iron. For more than about five years no one used water from the well. It was dug and water came, but then it was left unused and it dried. So, what happened then is that the rust from the pipe leached into the water." LMF 4

The participants' accounts indicate that water availability in the settlement is insufficient for all personal and domestic activities. As a result of lacking water to wash clothes and dishes, some of the movement's members reduced their bathing frequency.

> "No, there's no water. I don't wash clothes. It's been four months since I've washed any clothes. And we have to wash dishes. I'm not even washing dishes properly, just the inside of them. But I'm hopeful God will send rain and then I'll wash my dishes. There's no way to wash properly." LMF 8

> "I've already tried cutting back on showers. I took one shower last week. I showered with a bucket of this water. There wasn't any water". LMF 9

The problem of availability affects the physical accessibility of water as some families have to fetch water at their neighbours' house. From the participants, one elderly person travelled as much as 8 km to fetch water. It was observed that the quantity of water fetched is not always enough to cover the inhabitants' daily needs.

> "So, we have to get water at the neighbours' house". LMF 6

> "We have to go to JP's house to get drinking water. Every other day we go. We get two gallons to not take advantage of him too much". LMF 1

In this specific case—a household with 4 inhabitants—they fetched a total of 20 litres per day, equivalent to 5 litres/person/day to drink and cook. This is an insufficient quantity to cover basic needs.

In order to improve their access to water, some families are building a well. However, part of the installation requires a pump to bring water to the surface, which entails costs for electricity. Many families do not have the means to pay for this additional expense:

> "We'll be spending too much and we'll have an even more expensive electricity bill to pay". LMF 9

> "To get water we spend 400 reals per month (value paid in electricity)." LMF 1

The problem of availability also results in the use of sources with questionable water quality. Some families filter water while others drink it without any treatment, leading to a variety of health risks.

> "When you get it from the pump and boil it, a sort of yellow cream forms on top, which means that somethings in that water. So, when you take that out, praise God, the water is clear after." LMF 8

"No, I don't have a filter and I don't boil [water] either. After it's in the cistern we just take it normally." LMF 7

In addition to this, water considered good by some is considered bad by others. In this sense, there is a lack of guidance and even of monitoring of water quality consumed in the settlement.

"Frankly, I do not drink this water. I only drink water from H. and we drink it filtered. I always filter water before drinking it. There are other people that drink it without filtration or anything". LMF 1

"There, beside H.'s house, the water there is very, very salty and on occasion we've had kidney problems. And over here, no. I don't know if we've already become used to the worms." LMF 3

In addition, the recipients used to fetch water are not always clean and properly sanitized, which contributes to deteriorating the quality of the water consumed.

The search for new sources of water generates insecurity and some families will not accept to drink water from an unknown source. They, moreover, do not grow used to bottled or even boiled water.

"We aren't drinking it because it smells of swamp. You know, that smell of rotten grass?" LMF 1

"I've got a problem with this water. I go into the street and drink water and then I want nothing else but to go drink my water full of germs. And I can't stand it [bottled water], it's as if my organism won't accept it. It's horrible, horrible. It's got this taste of, like, treatment, they're overzealous when they put it in. I asked for some water and spit it right out. And I said, it's horrible. I don't even like boiling water, it's horrible. It would be better for me to boil it, but we just filter it." LMF 6

Regarding sanitation, some families do not have a toilet in the household, while others do have one but lack water to flush. The lack of available water hence infringes on the right to sanitation as it increases the rate of open defecation.

"I go into the brush close by where it floods and where there's lots of [tall grass]. Then, when it's going to rain, the water takes it all away because I relieve myself there. There is a toilet, the bowl is there and I keep it all clean. There's a wall around it and I covered it all in plastic, all organized like. And I pour water in the bowl and it's all clean. But there's no water, how am I supposed to use it?" LMF 8

## Water as a tool for food production for self-subsistence and income generation

As the Landless Movement workers make their living from agricultural production, their production is necessary for their own consumption (to prevent malnutrition) and for sale to generate income to buy other foodstuffs and products required for their survival. Insufficient water affects production and hampers access to food and health promotion for this social group. It is worth highlighting that MST is a relevant stakeholder in the fight against the use of pesticides. The movement is a proponent of producing healthy foods through agroecology, which has specific production-related requirements.

"Because I came here to work the land and grow produce, and without water there's no way for us to produce anything. Beans were growing quite well and now we can't plant or do anything. There's no rain. And here there's no water to water the plants. Even my garden, which was so pretty. It's all over now." LMF 1

"We have our little garden there and, when it would rain, it was working out. But now. . . It would fill up our jugs and give enough to douse everything properly. Now everything's drying up. Now there are just seeds, but I'll pick the seeds and keep them. When it rains, I'll plant them again. I would pick broccoli, kale, green onion. Lots of stuff." LMF 3

"There's no water for us to be planting anything. The greatest difficulty, I think, at least on my plot is water, because we want to plant". LMF 3

In addition to agricultural production, the lack of water hampers the ability to raise livestock and produce milk. Due to the scarcity, the families are forced to seek other sources of income.

"We're tired of saying that an animal can't live without water. Without grazing land, it's not even that bad. But, unfortunately, without water there's no way to survive. That's why, lately, I don't have any livestock". LMF 2

"With the lack of water, whatever I had started slipping away. There's no more milk. It's been about three years that I don't even have a garden because when I do and it starts to sprout, it dies. And other times it doesn't even sprout." LMF 6

For many families access to water must urgently be improved in the region in order for them to continue planting crops. If not, they will have to search for another place to live.

"If all the water goes away there's no way. Then all our work, all our lives that we had here, all fades away." LMF MST 3

## Discussion

The research demonstrates that the problem of lacking availability of water in rural regions, in addition to being a catalyst to violate other elements of the normative content of the HRWS, impacts on other human rights.

Concerning the normative content of the HRWS, it is clear that lacking availability entails the need to fetch water in more distant places and from sources that are not safe, which impacts on the quantity of water that families use. Often, when fetching water, families are unable to obtain the necessary quantity for basic activities owing to the weight of the water buckets and the travel time required. Beyond affecting the quantity of water consumed, this situation implies that some families members—in particular women—have to make trips to fetch water, which hampers the HRWS normative content related to accessibility. Other families have to use a pump to obtain water from wells, the costs of which increase the amounts paid by the settlers for electricity. In this way, the settled families end up paying a higher amount for electricity in order to have access to water, which hinders affordability. Quality of water is also compromised as, in some cases, families recur to alternative sources of water. Some families reported difficulties with the smell and taste of water that they drink, thus indicating a hindrance to acceptability [2, 3].

As highlighted, beyond the effects on other elements of the normative content, limited availability of water interferes with food production activities for self-subsistence and income generation. Therefore, water and food sovereignty should be considered as an integrated concept in the rural context. Unsustainable water sovereignty undermines not only food sovereignty but also activities related to cultural processes and health. For instance, the growth and use of medicinal plants, which require water availability, is a caretaking practice that is part of countryside life and the community's traditional heritage [21]. Indeed, the countryside population's identity incorporates traditional knowledge as an intimate relationship with nature and growth of products destined for commercialization and own consumption. Thus, difficulty in working the land can obstruct MST members' ability to build an identity, with consequent impacts on their lifestyles and health [16, 22–25]. In this way, the consequences of lacking availability of water, in terms of quantity and quality, to at least meet the settlers' basic needs can entail a forced exodus from the countryside. This would not be the case if adequate infrastructure existed for these activities [17].

Quality of life for rural communities is fulfilled, to a large extent, through adequate access to land with guaranteed enjoyment of the human rights to water and sanitation. The adequate access to water creates the possibility, for such populations, to build relationships and to strengthen their specific ways of life. Land ensures income, food for self-subsistence and the growth of goods that are essential for the inhabitants' health, but requires that the community has access to the resources that it can provide [7, 9–11, 16].

Land distribution in developing countries is a vital tool to reduce food insecurity, poverty and inequalities, generating opportunity and providing the freedom to small farmers [25]. The search for social justice hinges, among other issues, on ensuring rights to access land for marginalized groups [7, 26]. However, for this group to be able to use the land and gain the resulting autonomy and social justice, the concept of access to land has to be expanded to include adequate access to basic resources, such as water.

Substantial challenges loom over the future of the community: the possibility of a rural exodus due to lacking water and the absence of public measures to ensure this good. A lack of public policies providing effective protection for this population can compromise their ability to remain in the region and to share knowledge about the countryside and all related sociocultural paraphernalia [15, 27].

In order to go on in the settlement, the families with the greatest difficulties in accessing water depend on neighbours. Social support is an important mechanism to deal with adversity and individuals in the encampment find support in their social networks to deal with lacking water. In this way, strong social cohesion is needed in these communities in order for them to be resilient [28]. Collective action, thus, becomes a source of hope and optimism that reinforces the importance of community action, such as when collective health practices or health promotion measures are carried out, especially in marginalized environments. It is also important to highlight that the community where this investigation took place makes sure that daily routines are harmonious with organizational values linked to MST principles. According to these principles, landless rural workers should embody an identity characterized by the struggle and resistance against rural exodus. They should ensure that agriculture carried out in their territories promotes them as sociable spaces. Food produced should be healthy and water, soil and biodiversity should be preserved [29]. Since water plays a central part in community development, uncertainty regarding its availability makes country life more precarious.

It can, thus, be appreciated that access to water is an important factor for life and, for rural communities, it is vital to self-subsistence and realization of the rights to food, human dignity and an adequate standard of living. The settlers depend on their crops and livestock to obtain adequate foodstuffs, which requires access to water. Competition often exists in rural areas

regarding the use of water and is brokered according to power relations. This reinforces the need to establish priorities to ensure that water is used foremost to ensure the basic needs of vulnerable populations [5, 6, 9–11, 26, 30].

Extreme poverty is associated with locations where access to water is uncertain; two-thirds of the world's population with malnutrition live in rural regions and depend on self-subsistence agriculture for their own food production and income generation [13]. Thus, the impact of lacking access to water in rural communities can entail food insecurity [28, 29]. Some studies even indicate that when communities lack access to water and have associated food insecurity and poverty, individuals' vulnerability increases and significant associations exist with the emergence of mental illnesses such as depression and anxiety [31–34].

Some authors have pointed out the need to understand access to water as a necessary condition for quality of life, to improve people's skillsets and broaden their opportunities to gain the autonomy required to live the life they want in freedom [5, 6, 8, 12, 26, 35–38]. Access to water in rural areas is often conditional upon power games that exclude some social groups based on economic, social or ethnic characteristics, among others. At the same time, access to water should be understood from a relational perspective that allows different communities to carry out the cultural practices that are part of their identity. Water, for some communities, is an essential element for their existence as a community, being essential for them to exercise their ancestral practices. It is not only to be used, but to be revered, taking care of the water cycle and its relationship with a safe environment and ecosystem protection. Seen in this light, communities' inability to obtain access to water translates into their loss of autonomy, identity and freedom. In this sense, allowing these social groups access to water would enable them to fully exercise their capabilities, including those essential for their survival, besides transforming social and power relations [5, 6, 9, 13, 27, 36–39].

Seeing access to water as a tool capable of producing liberty in a community requires acknowledgment of the fact that access o water cannot be understood in purely quantitative terms, reduced to matters of having water to drink and perform personal and domestic hygiene. Water is a prerequisite to end poverty and improve quality of life. Thus, the HRWS must be ensured in a multidimensional sphere. Recognizing water as a resource that determines peoples' lifestyles and can impinge on their quality of life is essential to the pursuit of an egalitarian society [5, 6, 8, 12, 26, 35–38].

Article 5 of the Vienna Declaration, adopted at the World Conference of Human rights in 2003, states that human rights are indivisible, interdependent and interrelated. In an assessment of access to water for homeless populations, for instance, it was shown that lack of access to water impacted on the human rights to education, work and leisure [39]. A similar situation is observed for the group analyzed in the present research as non-enjoyment of the HRWS also obstructed the associated population's rights to food, health and an adequate standard of living. Therefore, access to water must be understood as a human right to be ensured for all individuals, without discrimination, in order for other human rights to also be realized [6, 26, 30]. In this sense, it is important to recall that States who signed and ratified the International Convention on Economic, Cultural and Social Rights have a legal obligation to respect, protect and fulfill those human rights, including the HWRS.

For rural workers, water represents life. The everyday activities carried out in their territories, such as agriculture, livestock raising and healthcare practises, depend on water. Thus, a lack thereof obstructs their access to food, self-care, hygiene, farming of medicinal plants, land management and symbolic interrelations that unfold in their territories. For that reason, social movements in Brazil's countryside demand the opportunity to have a land to live in, with adequate access to water, mobilizing their traditional and politico-social knowledge and bolstering their social identity, a characteristic of the values and principles that they uphold [16, 22]. For this to become a reality, access to water framed by the human rights framework is required.

The findings of this research, reinforces the idea that the HWRS can be a reference framework for public policies covering access to water and sanitation geared to rural populations. Those public policies, once based on human rights principles, can reduce social inequalities and situations of vulnerability while also promoting health [40, 1]. Importantly, it must be recalled that this population's use of water is not limited to drinking, cooking and ensuring hygiene in the household. Cultural and environmental specificities modify how a community uses water, while asymmetrical power relations exercise an influence on access. Thus, a holistic and multidimensional vision of water uses, encompassing the cultural, social and political relations that permeate access to water, must be considered during the formulation of policies, programs and measures aimed at achieving universal, equal access.

Important to say that this study, as all researches, has some limitations. It is necessary to consider that the findings came from a qualitative research, which was based, mainly, on what was reported by the participants living at Ulisses Oliveira settlement. Therefore, it is more indicative than representative of the situation, and we suggest that other researches should be done in other rural settlement to see if similar findings come out. Notwithstanding, valuable lessons could be drawn from the findings in this paper.

## Conclusion

This research highlights that access to water is a fundamental factor to ensure life conditions with dignity for rural populations. Upon consideration of access to water in an MST settlement through the lens of the HRWS, it was verified that lacking availability impacts on all other elements of the normative content. It also hinders other rights, particularly the right to food, which can entail food shortages, in addition to impinging on cultural activities that also impact on quality of life. In sum, when analyzing the HRWS with consideration for rural populations, in addition to the need for water for the purposes of drinking, cooking and ensuring personal and domestic hygiene, the need to acknowledge access to water for food production is required. Indeed, understanding the characteristics of each context is essential to develop policies that aim to minimize the problems resulting from a lack of access to water and sanitation.

Furthermore, understanding of the specificities of the access to water and sanitation for rural populations and the development of public policies able to ensure this access can be a factor for reducing rural exodus, which is not only a problem for the rural communities but a social problem with a substantial social and economic cost.

Thus, access to water must be understood in the light of its political, social and cultural dimensions. Moreover, public policies must also be multidimensional, aiming to improve quality of life for rural populations and to ensure the HRWS.

## Supporting information

**S1 File. Research script.**
(PDF)

**S2 File. Aproval of the ethical committee.**
(PDF)

## Acknowledgments

We want to acknowledge Lucinha and Mr. Chico for receiving us at their houme, and everyone that lives at Ulisses Oliveira settlement. We also want to thank the two reviewers, whose comments significantly contributed to the improvement of this paper.

## Author Contributions

**Conceptualization:** Priscila Neves-Silva, Léo Heller.

**Data curation:** Priscila Neves-Silva, Juliana Aurora de Oliveira Lopes.

**Formal analysis:** Priscila Neves-Silva, Juliana Aurora de Oliveira Lopes.

**Funding acquisition:** Léo Heller.

**Investigation:** Priscila Neves-Silva, Juliana Aurora de Oliveira Lopes.

**Methodology:** Priscila Neves-Silva.

**Project administration:** Priscila Neves-Silva.

**Supervision:** Léo Heller.

**Validation:** Léo Heller.

**Writing – original draft:** Priscila Neves-Silva, Juliana Aurora de Oliveira Lopes.

**Writing – review & editing:** Léo Heller.

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
