## [Decision Letter · Decision Letter 0]

22 Apr 2020

PONE-D-20-04524

The right to water: impact on the quality of life of rural workers in a settlement of the Landless Workers Movement, Brazil

PLOS ONE

Dear Mrs Neves-Silva,

Thank you for submitting your manuscript to PLOS ONE. After careful consideration, we feel that it has merit but does not fully meet PLOS ONE’s publication criteria as it currently stands. Therefore, we invite you to submit a revised version of the manuscript that addresses the points raised during the review process.

We would appreciate receiving your revised manuscript by Jun 06 2020 11:59PM. To enhance the reproducibility of your results, we recommend that if applicable you deposit your laboratory protocols in protocols.io, where a protocol can be assigned its own identifier (DOI) such that it can be cited independently in the future. For instructions see: http://journals.plos.org/plosone/s/submission-guidelines#loc-laboratory-protocols

We look forward to receiving your revised manuscript.

Kind regards,

Ho Ting Wong, PhD

Academic Editor

PLOS ONE

Journal Requirements:

2. When reporting the results of qualitative research, we suggest consulting the COREQ guidelines: http://intqhc.oxfordjournals.org/content/19/6/349. In this case, please consider including more information on the number of interviewers, their training and characteristics; and if bias issues were considered. Moreover, please provide the script used in the interviews; and provide a table describing the demographic characteristics of the participants.

Additional Editor Comments (if provided):

I recognize that there are typos on page 2 line 54, and page 3 line 68 (the superscript 2 and 3).

On the other hand, one of the reviewers suggested that the English standard of your manuscript should be improved. Please think about what can be done to improve.

Reviewers' comments:

Reviewer's Responses to Questions

**Comments to the Author**

1. Is the manuscript technically sound, and do the data support the conclusions?

Reviewer #1: Yes

Reviewer #2: Yes

2. Has the statistical analysis been performed appropriately and rigorously? 

Reviewer #1: I Don't Know

Reviewer #2: N/A

3. Have the authors made all data underlying the findings in their manuscript fully available?

Reviewer #1: Yes

Reviewer #2: Yes

4. Is the manuscript presented in an intelligible fashion and written in standard English?

Reviewer #1: No

Reviewer #2: Yes

5. Review Comments to the Author

Reviewer #1: The manuscript "The right to water: impact on the quality of life of rural workers in a settlement of the

Landless Workers Movement, Brazil" provides an important analysis of water availability, access and control within the context of a settlement of Brazil's Landless Workers' Movement. This is a timely article, as there has been a paucity of scholarship on the right to water within the MST. While excellent, I believe that with moderate revisions the manuscript will be better positioned to make a contribution to the literature. First, the article is in need of greater theoretical engagement. Many of the quotes that the author(s) present are lacking in analysis. There should be greater theoretical analyses of the results, and these should be tied to the theoretical framework that is presented in the beginning of the manuscript. As part of this revision, please ensure that this paper is making an explicit contribution to theory; at present the contribution is unclear. One possibility would be to make a stronger theoretical connection to debates surrounding questions of sovereignty. The authors bring up food sovereignty, and I believe in this context it would be helpful to offer an analysis of how sovereignty over water, like over food or energy, are intertwined struggles within La Via Campesina, and how these various forms of sovereignty, and the threats to them, overlap, so that problems of water sovereignty overlap and condition food sovereignty, for example. The last comment is stylistic; while the article is exceptionally well-written, it would benefit from the careful attention of a native English speaker to ensure that the prose and arguments made are accessible.

Reviewer #2: I consider that the manuscript worths to be published. Notwithstanding, I as well consider that the following remarks should be taken into account by authors to proceed a revision on the paper, as a condition for publishing it.

There is given lot of relevance for United Nations declaration, as if from this declaration on, everybody magically has acess to water. I would like to suggest this tone to be a little bit changed.

In rural areas, water is used in a much higher quantity for agriculture rather than to be used by people. Could not agree on the expression "water is also often vital for food production". Access to water in rural areas is a duty for the State in exception situations only.

Line 83 should make reference to the 1966 Pact on Economic, Social and Cultural rights.

Line 85. I did not understand if the phrase is said by the authors of the manuscript or by Metha. Anyway, I would like to know why it wolud be impossible to separate household uses of water from productive ones.

Line 91. Is HRWS free of charge? Would people be allowed to use (and abuse) the right to water? Neither responsibility on sustainability measures nor simply a duty to pay for the services of having treated water at home?

The point, by and large, is that I did not see much relation between this introduction (from line 1 to 98) and the subject of the paper. Why the authors express so much about the International Right to Water (and many interpretations to it)? In line 88 it is said that the HRWS is "a framework of reference". How? And what is that for?

Lines 110-114 -- It would be better to say that as a hypothesis rather than as a conclusion.

Lines 122-155 -- Interviews with people should be authorised by an Ethic Committe, indicated at lin 182. I request the authors to present that authorisation to Editorial Board of the Journal as a condition for publishing the paper.

Lines 281 and 390-- I wonder if "malnutrition" could be substituted for "widespread hunger" or "starving"

Line 313 -- "be improved" instead of "improve" sounds better.

Line 325 -- "as safe" should be substituted for "safe"

Lines 328-9 -- wording should be improved

Lines 339-341. I haven't understood the relation authors want do do between caretaking practice and the right to water or the lack of water in the region. I ask authors to be more precise on it.

Line 352 -- Although it is written that "land ensures income", the paper says that it does not. Land is necessary but not sufficient (as correctly said in line 359).

Line 374 -- I guess there is something wrong here in this expression: "community undergoing investigation"

Line 397 -- Which authors have pointed that out?

Line 402 -- Why it should be understood this way? I would like to see authors' arguments on it.

Line 415 -- There as International Documents that say exactly this. See, for instance, the Vienna Declaration, adopted at the World Conference of Human Rights (2003), mainly article 5 (but feel free to quote from the first "consideranda" to articles 1-5)

Line 423 -- Do States really have any legal obligation to respect, protect and fulfill human rights? There is a lot of discussion on it. Anyway, legal obligation really starts when a State internally recognises a Human Right by positing a law by which that right shall be protected. For instance, the right to health in Brasil is respected in the terms of Law 8080 (SUS), rigth to housing in the terms of Minha Casa Minha Vida Program and so on...

Line 429 -- I did not understand "right to live on land, but also the right to land" From my point of view, in both cases is possible to fight for access to water, and in a very equivalent manner.

Lines 432-434. I didn't understand. "The use of human rights...can reduce social inequalities"? I believe that human rights can be the grounding for a public policy, but it is the publicy policy, duly implemented, the instrument that actually can reduce social inequalities. Human rights on their own cannot magically cause reduction of inequalities.

Line 449. It is clear that access to water is a fundamental factor to ensure life conditions, not only life with dignity!

In Conclusions, considering that authors want to demand for a public policy, they should at least say that rural exodus is a problem that is expensive and that the cost for keeping people in rural areas (furnishing water to them) is less expensive. You can only propose a public policy if Government understands that there is a lack of rights (and there is a lack of right to water) AND a social problem that is bigger and that need to be avoided. On the agenda setting of public policies, cf. John Kingdon or, in Portuguese, Celina Sousa.

6. PLOS authors have the option to publish the peer review history of their article (what does this mean?). If published, this will include your full peer review and any attached files.

Reviewer #1: No

Reviewer #2: Yes: Josué Mastrodi

---

## [Author Response · Author response to Decision Letter 0]

4 Jun 2020

Belo Horizonte, May 26, 2020

To: PloS One Journal

Dear Editor,

Prof. Ho Ting Wong

Ref. PONE-D-20-04524 – “The right to water: impact on the quality of life of rural workers in a settlement of the Landless Workers Movement, Brazil"

First of all we would like to thank you and the two reviewers of our manuscript for the contributions provided that certainly were instrumental for its improvement. They were all valued and we tried to accomodate most of them. Please, see bellow our comments to all recommendations of your letter.

Journal Requirements:

Answer: The requirements was observed and modified as asked.

2. When reporting the results of qualitative research, we suggest consulting the COREQ guidelines: http://intqhc.oxfordjournals.org/content/19/6/349. In this case, please consider including more information on the number of interviewers, their training and characteristics; and if bias issues were considered. Moreover, please provide the script used in the interviews; and provide a table describing the demographic characteristics of the participants.

Answer: The information about the number of interviewers was outlined in line 144. The script was sent separately as supplementary material and the demographic characteristics of the participants are described in line 144 to 146.

Answer: We confirm that data from this study are available upon request. This is due to ethical reasons. In the Informed Written Consent, signed by the participants, we affirm that no one will access the data collected, apart from the investigators. This is a necessary precaution to preserve the participant's anonymity, since it is possible for someone from the community to deduce from the full transcripts the identity of the participant. This was a request made by the ethic committee in order to approve the investigation. The ethic committee contact is cepcoord.minas@fiocruz.br and the first author’s contact is priscila.neves31@gmail.com.

Additional Editor Comments:

Question: I recognize that there are typos on page 2 line 54, and page 3 line 68 (the superscript 2 and 3).

Answer: Thank you, we corrected them 

Question: On the other hand, one of the reviewers suggested that the English standard of your manuscript should be improved. Please think about what can be done to improve.

Answer: We undertook a thorough review of the language of the manuscript.

Reviewer #1: 

Question: First, the article is in need of greater theoretical engagement. Many of the quotes that the author(s) present are lacking in analysis. There should be greater theoretical analyses of the results, and these should be tied to the theoretical framework that is presented in the beginning of the manuscript. 

Answer: We appreciate this comment and worked to improve the links between the theoretical framework and the analysis. For this, we have attempted to improved the section Discussion and tied it in a more precise way to the theoretical framework presented in the Introduction, mainly the HRWS. 

Question: As part of this revision, please ensure that this paper is making an explicit contribution to theory; at present the contribution is unclear. One possibility would be to make a stronger theoretical connection to debates surrounding questions of sovereignty. The authors bring up food sovereignty, and I believe in this context it would be helpful to offer an analysis of how sovereignty over water, like over food or energy, are intertwined struggles within La Via Campesina, and how these various forms of sovereignty, and the threats to them, overlap, so that problems of water sovereignty overlap and condition food sovereignty, for example. The last comment is stylistic; while the article is exceptionally well-written, it would benefit from the careful attention of a native English speaker to ensure that the prose and arguments made are accessible.

Answer: As asked, we have strengthened the debate on water and food sovereignty and decided to rearrange the two paragraphs that come aftter in order to see if the idea is clear. (please see lines 93 to 114 and 338 to 342)

Reviewer #2: 

Question: There is given lot of relevance for United Nations declaration, as if from this declaration on, everybody magically has access to water. I would like to suggest this tone to be a little bit changed.

Answer: We support the idea that the HRWS "can be used as a framework to regulate the access of water in several countries and have the potential of significantly contributing to public policy formulation and implementation". We introduced some changes in the text in order to made it clearer. Lines 32 to 39

Question: In rural areas, water is used in a much higher quantity for agriculture rather than to be used by people. Could not agree on the expression "water is also often vital for food production". Access to water in rural areas is a duty for the State in exception situations only.

Answer: We used a more nuanced language and tried to clarify our position on this issue (line 49)

Question: Line 83 should make reference to the 1966 Pact on Economic, Social and Cultural rights.

Answer: We included it, lines 67 and 68

Question: Line 85. I did not understand if the phrase is said by the authors of the manuscript or by Metha. Anyway, I would like to know why it would be impossible to separate household uses of water from productive ones.

Answer: We tried to improve the text and to clarify this issue (lines 69 to 77)

Question: Line 91. Is HRWS free of charge? Would people be allowed to use (and abuse) the right to water? Neither responsibility on sustainability measures nor simply a duty to pay for the services of having treated water at home?

Answer: We clarified this point in lines 82 to 87

Question: The point, by and large, is that I did not see much relation between this introduction (from line 1 to 98) and the subject of the paper. Why the authors express so much about the International Right to Water (and many interpretations to it)? In line 88 it is said that the HRWS is "a framework of reference". How? And what is that for?

Answer: We expanded the explanation about the connection of the framework of the HRWS and the approach of the manuscript. At the same time, we improved the Discussion session, articulating in a clearer way with the theoretical framework.

Question: Lines 110-114 -- It would be better to say that as a hypothesis rather than as a conclusion.

Answer: We decided to delete this paragraph and rearrange the two paragraphs before. After introducing what the first reviwer asked, abour soverenignty, we thought that would be better to rearrange those paragraphs.

Question: Lines 122-155 -- Interviews with people should be authorised by an Ethic Committe, indicated at line 182. I request the authors to present that authorisation to Editorial Board of the Journal as a condition for publishing the paper.

Answer: It was submitted separately as supplementary material

Question: Lines 281 and 390-- I wonder if "malnutrition" could be substituted for "widespread hunger" or "starving"

Answer: We believe that malnutrition is a more precise wording since the population produces food, but it needs to be in a sufficient quantity and multiple variety to provide them with a minimum nutritional level.

Question: Line 313 -- "be improved" instead of "improve" sounds better.

Answer: We modified it as asked. Line 314

Question: Line 325 -- "as safe" should be substituted for "safe"

Answer: We modified it as asked. Line 325

Question: Lines 328-9 -- wording should be improved

Answer: We improved this part as asked. Lines 328 to 331.

Question: Lines 339-341. I haven't understood the relation authors want to do between caretaking practice and the right to water or the lack of water in the region. I ask authors to be more precise on it.

Answer: We tried to improve the arguments. Lines 342 to 345

Question: Line 352 -- Although it is written that "land ensures income", the paper says that it does not. Land is necessary but not sufficient (as correctly said in line 359).

Answer: We modified the text accordingly. Lines 353 to 359

Question: Line 374 -- I guess there is something wrong here in this expression: "community undergoing investigation"

Answer: We modified it. Line 379

Question: Line 397 -- Which authors have pointed that out?

Answer: We included it. Line 405

Question: Line 402 -- Why it should be understood this way? I would like to see authors' arguments on it.

Answer: We tried to improve the arguments. Lines 409 to 412

Question: Line 415 -- There as International Documents that say exactly this. See, for instance, the Vienna Declaration, adopted at the World Conference of Human Rights (2003), mainly article 5 (but feel free to quote from the first "consideranda" to articles 1-5)

Answer: We included this information. Line 424

Question: Line 423 -- Do States really have any legal obligation to respect, protect and fulfill human rights? There is a lot of discussion on it. Anyway, legal obligation really starts when a State internally recognises a Human Right by positing a law by which that right shall be protected. For instance, the right to health in Brasil is respected in the terms of Law 8080 (SUS), rigth to housing in the terms of Minha Casa Minha Vida Program and so on...

Answer: We tried to improve the text and to clarify this issueanswered it in lines 432 to 435

Question: Line 429 -- I did not understand "right to live on land, but also the right to land" From my point of view, in both cases is possible to fight for access to water, and in a very equivalent manner.

Answer: We modified it in line 440 to 444

Question: Lines 432-434. I didn't understand. "The use of human rights...can reduce social inequalities"? I believe that human rights can be the grounding for a public policy, but it is the public policy, duly implemented, the instrument that actually can reduce social inequalities. Human rights on their own cannot magically cause reduction of inequalities.

Answer: We modified the text in lines 445 to 449

Question: Line 449. It is clear that access to water is a fundamental factor to ensure life conditions, not only life with dignity!

Answer: We modified the text in lines 463 to 464

Question: In Conclusions, considering that authors want to demand for a public policy, they should at least say that rural exodus is a problem that is expensive and that the cost for keeping people in rural areas (furnishing water to them) is less expensive. You can only propose a public policy if Government understands that there is a lack of rights (and there is a lack of right to water) AND a social problem that is bigger and that need to be avoided. On the agenda setting of public policies, cf. John Kingdon or, in Portuguese, Celina Sousa.

Answer: We improved the arguments in lines 474 to 477

Hope we could answer apropriatelly all your questions and improve the paper.

Sincerelly yours,

Priscila Neves-Silva

---

## [Decision Letter · Decision Letter 1]

26 Jun 2020

PONE-D-20-04524R1

The right to water: impact on the quality of life of rural workers in a settlement of the Landless Workers Movement, Brazil

PLOS ONE

Dear Dr. Neves-Silva,

Thank you for submitting your manuscript to PLOS ONE. After careful consideration, we feel that it has merit but does not fully meet PLOS ONE’s publication criteria as it currently stands. Therefore, we invite you to submit a revised version of the manuscript that addresses the points raised during the review process.

We look forward to receiving your revised manuscript.

Kind regards,

Ho Ting Wong, PhD

Academic Editor

PLOS ONE

Additional Editor Comments (if provided):

Regarding item 2 of the journal requirement, you are required to provide information related to the interviewers.

However, I find that the information in line 144 to 146 that you mentioned in your response are only related to the interviewees.

Reviewers' comments:

Reviewer's Responses to Questions

**Comments to the Author**

1. If the authors have adequately addressed your comments raised in a previous round of review and you feel that this manuscript is now acceptable for publication, you may indicate that here to bypass the “Comments to the Author” section, enter your conflict of interest statement in the “Confidential to Editor” section, and submit your "Accept" recommendation.

Reviewer #2: All comments have been addressed

2. Is the manuscript technically sound, and do the data support the conclusions?

Reviewer #2: Yes

3. Has the statistical analysis been performed appropriately and rigorously? 

Reviewer #2: N/A

4. Have the authors made all data underlying the findings in their manuscript fully available?

Reviewer #2: Yes

5. Is the manuscript presented in an intelligible fashion and written in standard English?

Reviewer #2: Yes

6. Review Comments to the Author

Reviewer #2: Considering that the suggested revisions were accepted in the ultimate version of the submitted article, I propose the article to be published.

7. PLOS authors have the option to publish the peer review history of their article (what does this mean?). If published, this will include your full peer review and any attached files.

Reviewer #2: **Yes: **Josué Mastrodi

---

## [Author Response · Author response to Decision Letter 1]

26 Jun 2020

To: PloS One Journal

Dear Editor,

Prof. Ho Ting Wong

Ref. PONE-D-20-04524 – “The right to water: impact on the quality of life of rural workers in a settlement of the Landless Workers Movement, Brazil"

Thank you and the reviewers for the contributions provided. Please, see bellow our comments to your request.

Additional Editor Comments:

Request: Regarding item 2 of the journal requirement, you are required to provide information related to the interviewers.

However, I find that the information in line 144 to 146 that you mentioned in your response are only related to the interviewees.

Answer: So sorry, I missed that request. I organized the information you asked, about interviewer characteristics and training, in lines 126-131. About consideration of bias issues, is was already described in lines 169-175. Script was sent with the last revision of the manuscript. Please, confirm if that is what you expected.

Reviewers' comments:

If I understood, about the reviewers, there is no further request. 

Hope I could answered your questions properly,

Best Regards,

Priscila Neves Silva

---

## [Editor Report · Decision Letter 2]

6 Jul 2020

The right to water: impact on the quality of life of rural workers in a settlement of the Landless Workers Movement, Brazil

PONE-D-20-04524R2

Dear Dr. Neves-Silva,

We’re pleased to inform you that your manuscript has been judged scientifically suitable for publication and will be formally accepted for publication once it meets all outstanding technical requirements.

Kind regards,

Ho Ting Wong, PhD

Academic Editor

PLOS ONE
---

## [Editor Report · Acceptance letter]

8 Jul 2020

PONE-D-20-04524R2 

The right to water: impact on the quality of life of rural workers in a settlement of the Landless Workers Movement, Brazil 

Dear Dr. Neves-Silva:

I'm pleased to inform you that your manuscript has been deemed suitable for publication in PLOS ONE. Congratulations! Your manuscript is now with our production department. 

Kind regards, 

on behalf of

Dr. Ho Ting Wong 

Academic Editor

PLOS ONE